# Brief Communication: Ice Sheet Elevation Measurements from the Sentinel-3A / 3B Tandem Phase

Malcolm McMillan[1], Alan Muir[2], Craig Donlon[3]

[1]UK Centre for Polar Observation & Modelling, Centre for Excellence in Environmental Data Science, Lancaster University, Lancaster, LA1 4YW, UK
[2]University College London, Gower Street, London, WC1E 6BT, UK
[3]European Space Agency, European Space Research and Technology Centre (ESA ESTEC), Keplerlaan 1, 2201 AZ Noordwijk, Netherlands

*Correspondence to*: Malcolm McMillan (m.mcmillan@lancaster.ac.uk)

**Abstract.** Over the coming decade, the quartet of Copernicus Sentinel-3 satellite altimeters will provide a continuous record of ice sheet elevation change. To ensure consistency of measurement between the four satellites, requires rigorous in-flight inter-comparison. To facilitate this, Sentinel-3B was initially flown in a unique tandem formation with Sentinel-3A, enabling near-instantaneous, co-located measurements of surface elevation to be acquired. Here, we analyse tandem measurements of ice sheet elevation, to show that both instruments operate with statistically equivalent accuracy and precision, even over complex ice margin terrain. This analysis demonstrates that both satellites can be used interchangeably to study ice sheet evolution.

## 1 Introduction

Long-term continuity of ice sheet elevation measurements is important for understanding the nature and drivers of ice sheet change (Sandberg Sørensen et al., 2018; Shepherd et al., 2019). Polar orbiting satellite radar altimeters have, for the past 30 years, provided such a record, and with it new insight into surface topography, ice mass loss, dynamical instabilities, surface processes and subglacial hydrology (Helm et al., 2014; Konrad et al., 2017, 2018; McMillan et al., 2013, 2016, 2019; Slater et al., 2018). The most recent satellites contributing to this record are the Sentinel-3 series (Donlon et al., 2014), which provides delay-Doppler altimetry measurements up to a latitude of 81.35°, with an on-the-ground revisit time of 27 days. Unlike previous polar altimetry missions, Sentinel-3 is part of the operational EU Copernicus Programme. As such Sentinel-3 is composed of four satellites, which together will deliver unbroken coverage until at least the end of this decade. Because of this unique configuration, it is vital that measurements from the four satellites (Sentinel-3A, B, C and D), are systematically compared in-flight, to determine whether their associated streams of data can be treated interchangeably by the scientific and service user communities. This is important not only for long term continuity, but also for optimising the use of these data when more than one satellite is operating simultaneously. For example, at a latitude of 75°, adjacent tracks of a single satellite are separated by

approximately 23 km, whereas when two identical satellites are flying in their nominal orbits, the across track separation decreases to around 11.5 km.

The first two Sentinel-3 satellites (-3A and -3B), were launched on 16[th] February 2016 and 25[th] April 2018, respectively. To facilitate the inter-comparison of these satellites, Sentinel-3B was initially placed into a 'Tandem' formation with Sentinel-3A, whereby both satellites followed the same ground track (within the across-track control range of ±1 km) with a 30 seconds separation (Clerc et al., 2020). This configuration was maintained between 7[th] June – 16[th] October 2018, so as to acquire three full cycles of delay-Doppler measurements in this tandem formation. Over Earth's ice sheets, these measurements are important because they provide contemporaneous (within 30 seconds), co-located and co-orientated (i.e. same track heading and footprint orientation) observations. Such a configuration allows a more robust inter-comparison than is normally possible, because it avoids many of the common challenges associated with instrument inter-comparison, by removing the confounding effects of surface backscattering anisotropy (Armitage et al., 2014), and any spatial or temporal changes in elevation. In this study, we utilise this unique dataset to perform the first systematic inter-comparison of Sentinel-3A and Sentinel-3B (S3A and S3B, respectively) tandem altimetry measurements over ice sheets, and to assess the extent to which these measurements can be used interchangeably by the glaciological community. Specifically, we analyse (1) the consistency of S3A and S3B radar echoes acquired over the entire Antarctic Ice Sheet, (2) the precision of the S3A and S3B instruments over Lake Vostok, and (3) the accuracy of S3A and S3B elevation measurements as compared to independent reference datasets.

## 2 Data & Study Sites

We analysed tandem phase Sentinel-3A and Sentinel-3B SRAL data that were acquired during the summer of 2018; using the Level-2 enhanced data product (ESA Product Baseline 2.27; part of the Processing Baseline Collection 003). Our assessment focused on both continental-scale analysis (Section 3), and more targeted assessment at three study sites in East Antarctica (Sections 4 & 5); the Lake Vostok and Dome C sites, which exhibit relatively low slope topography that is characteristic of the ice sheet interior, and a coastal region of Wilkes Land (135-147°E, 66-70°S) which presents steeper and less uniform topography (McMillan et al., 2019). To assess the accuracy of the Sentinel-3A and Sentinel-3B elevation measurements at our study sites, we used airborne reference data acquired by the Airborne Topographic Mapper (ATM) and Riegl Laser Altimeter (RLA) instruments carried on Operation IceBridge campaigns (Blankenship et al., 2012; Studinger, 2014). Further details of these datasets and the method of inter-comparison are given in McMillan et al., 2019.

## 3 Consistency of delay-Doppler echoes acquired over ice sheets

When radar altimeters overfly areas of complex surface topography, the returned echo diverges from its classical shape (Ray et al., 2015) and the range to the altimeter can also change rapidly. These effects can complicate the reliable retrieval of surface

elevation information, because they can induce distortions in the theoretical waveform shape, impact upon the multi-looking process, and lead to multiple superimposed reflections from distinct surfaces within the doppler beam footprint. Handling these complex echoes is one of the major challenges associated with processing delay-Doppler altimetry data over regions of complex topography; affecting both the retracking process and also the Doppler beam stacking employed by altimeters such as Sentinel-3. To evaluate the impacts of complex topography upon delay-Doppler altimeter measurements, we therefore used the tandem phase to investigate the consistency of simultaneously acquired S3A and S3B waveforms over the entire Antarctic Ice Sheet. This analysis was motivated in part by the desire to determine whether (1) the complex waveform shape that is often apparent in coastal regions is essentially non-repeatable due to the pseudo-random combination of multiple reflections from within the Doppler beam footprint, or (2) whether this waveform complexity is repeatable, and therefore represents meaningful geophysical information about the surface geometry. This distinction is important, because the former implies that the signal is somewhat degraded; particularly when it comes to making stable, repeatable measurements through time. Whereas the latter implies that, whilst more sophisticated processing may be required, there is useable, physically meaningful information encoded within the complex waveform shape. To investigate this question, we therefore analysed tandem acquisitions over the entirety of Antarctica for one complete orbit cycle, comprising ~ 4 million S3A-S3B co-located and co-orientated waveform pairs. To quantify the similarity of each pair, we first aligned the waveforms within the range window according to their centre of gravity, and then computed the Pearson correlation coefficient, R, between each waveform pair (Figure 1). Finally, we averaged these measurements on a regular 5 by 5 km grid to investigate the extent to which the correlation coefficient varied as a function of ice sheet surface slope (Figure 1).

At the ice sheet scale, we find a very high level of agreement between S3A and S3B waveforms, with 92% of all waveforms having a correlation coefficient greater than 0.9. Importantly, high levels of correlation are not only limited to the relatively smooth interior of the ice sheet, but are also common across much of the ice sheet margin, which presents steeper and more complex topography. In these cases, we find that although the altimeter waveforms display a high degree of complexity, often with multiple peaks and a varying shape, the S3A and S3B waveforms still maintain their coherency, both in terms of their shape (e.g. number of distinct peaks), and the amplitude of the backscattered signal (Figure 1). This suggests that meaningful, repeatable information is encoded within complex waveform morphology, opening up the future possibility of utilising the full waveform to retrieve additional topographic information (e.g. through the use of an auxiliary Digital Elevation Model). Finally, we evaluated the relationship between surface slope and S3A-S3B waveform consistency by comparing the average surface slope (Slater et al., 2019) and average correlation coefficient, within $0.2^{\circ}$ slope intervals (Figure 1b). This clearly demonstrates the impact of surface slope upon waveform repeatability; namely that R decreases with increasing slope. Nonetheless the reduction in the correlation coefficient, R, is relatively modest, with the mean R decreasing from R > 0.9 (slopes lower than $0.6^{\circ}$), to R > 0.8 (slopes up to $1^{\circ}$), to R > 0.7 (slopes up to $2^{\circ}$).

## 4 Assessment of Instrument Precision at Lake Vostok

Next, we assessed and inter-compared the precision of the Sentinel-3A and Sentinel-3B altimeters by evaluating repeated elevation profiles that crossed the ice surface above subglacial Lake Vostok (Figure 2). This site provides a stable, relatively smooth (at the footprint scale) and low-slope surface that is well established for validation studies (McMillan et al., 2019; Richter et al., 2014). We selected a track that crossed above the central part of the subglacial lake and, for each satellite, we accumulated consecutive cycles acquired during the tandem phase of operations (S3A cycles 34-36; S3B cycles 11-13). Inspecting these data, we find no discernible difference between the measurements made by each satellite (Figure 2). To quantify the precision of both instruments, and to determine whether there was a statistically significant difference in their performance, we computed the standard deviation of all measurements made by each satellite within 1 km intervals along the satellite track. We used the estimated standard deviation as a measure of the precision of elevation measurements along the satellite track (Figure 2), which averaged 0.09 m and 0.10 m for S3A and S3B, respectively. Testing for significance (5% significance threshold) using the non-parametric Mann Whitney U (Hollander et al., 2015) and Kolmogorov-Smirnov (Massey, 1951) tests for the central values and distribution, respectively, we find that there is no significant difference in the instrument precision of Sentinel-3A and Sentinel-3B.

## 5 Elevation Accuracy

Finally, we assessed the absolute accuracy of Sentinel-3A and Sentinel-3B ice sheet measurements, by computing elevation differences relative to the Operation IceBridge reference datasets described in Section 2, using the approach described in McMillan *et al.*, 2019. We performed the analysis at three different sites (two inland sites, Lake Vostok and Dome C; and one coastal site, Wilkes Land) and using the two different retrackers provided in the ESA Level-2 product (the 'ice margin' retracker and the Threshold Centre of Gravity (TCOG) retracker). Across all sites and retrackers, we find that the differences in accuracy between S3A and S3B are always insignificant (5% significance level; using the same statistical tests described in Section 4), both in terms of the absolute biases relative to the reference datasets, and also the dispersion of the elevation differences (where the dispersion is defined as the median absolute deviation from the median; Figure 3). Using the TCOG retracker, for example, and comparing the S3A and S3B biases and dispersions, we find absolute differences between the two sensors of 2 mm (difference in bias) and 0.01 m (difference in dispersion) at our inland sites. At Wilkes Land, these absolute differences between S3A and S3B biases and dispersions increase to 0.19 m (bias) and 0.64 m (dispersion), but are still statistically insignificant. For context, the absolute bias and dispersions of each sensor relative to the reference data are of the order of 1-10 cm's at our inland sites, and of the order of 1 metre at our Wilkes Land site. This analysis therefore indicates that there is no significant difference in the accuracy of the two instruments across any of the sites studied, and that it is reasonable, from an instrument fidelity perspective, to use data from both satellites interchangeably.

## 6 Conclusion

This Brief Communication summarises the first detailed analysis of Sentinel-3A/B tandem phase measurements of ice sheet elevation. We find that (1) there is no significant difference between S3A and S3B instrument precision, (2) that there is no significant difference between the accuracy of S3A and S3B elevation measurements, and (3) that there is a high degree of correlation in co-located waveforms acquired by both instruments, even over complex coastal terrain. This study demonstrates that both satellites can be used interchangeably to monitor ongoing ice sheet evolution; effectively doubling the spatial coverage of measurements available, now that Sentinel-3B has moved to its nominal orbit. More broadly, it also establishes the value of operating a tandem phase immediately after satellite launch, and demonstrates that such operations should be performed when the Sentinel-3C and Sentinel-3D units enter service in the future.

**Data Availability**

The Sentinel-3 altimetry data used in this study are freely available through the Copernicus Open Access Hub (https://scihub.copernicus.eu/dhus/#/home). The IceBridge airborne altimetry data used in this study are freely available from the US National Snow and Ice Data Center (https://nsidc.org/).

**Author Contribution**

MM designed the experiments. MM and AM processed and analysed the data. MM prepared the manuscript with contributions from AS and CD, and all authors reviewed the manuscript.

**Competing Interests**

The authors declare that they have no conflict of interest.

**Acknowledgements**

This work was undertaken as part of the European Space Agency funded *Sentinel-3 Tandem for Climate* study (http://s3tandem.eu; Contract 4000124211/18/I-EF). The work was supported by the UK NERC Centre for Polar Observation and Modelling, the Sentinel-3 Mission Performance Centre, and the Lancaster University-UKCEH Centre of Excellence in Environmental Data Science. We thank the Editor and two anonymous reviewers for their comments, which have substantially improved the manuscript.

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

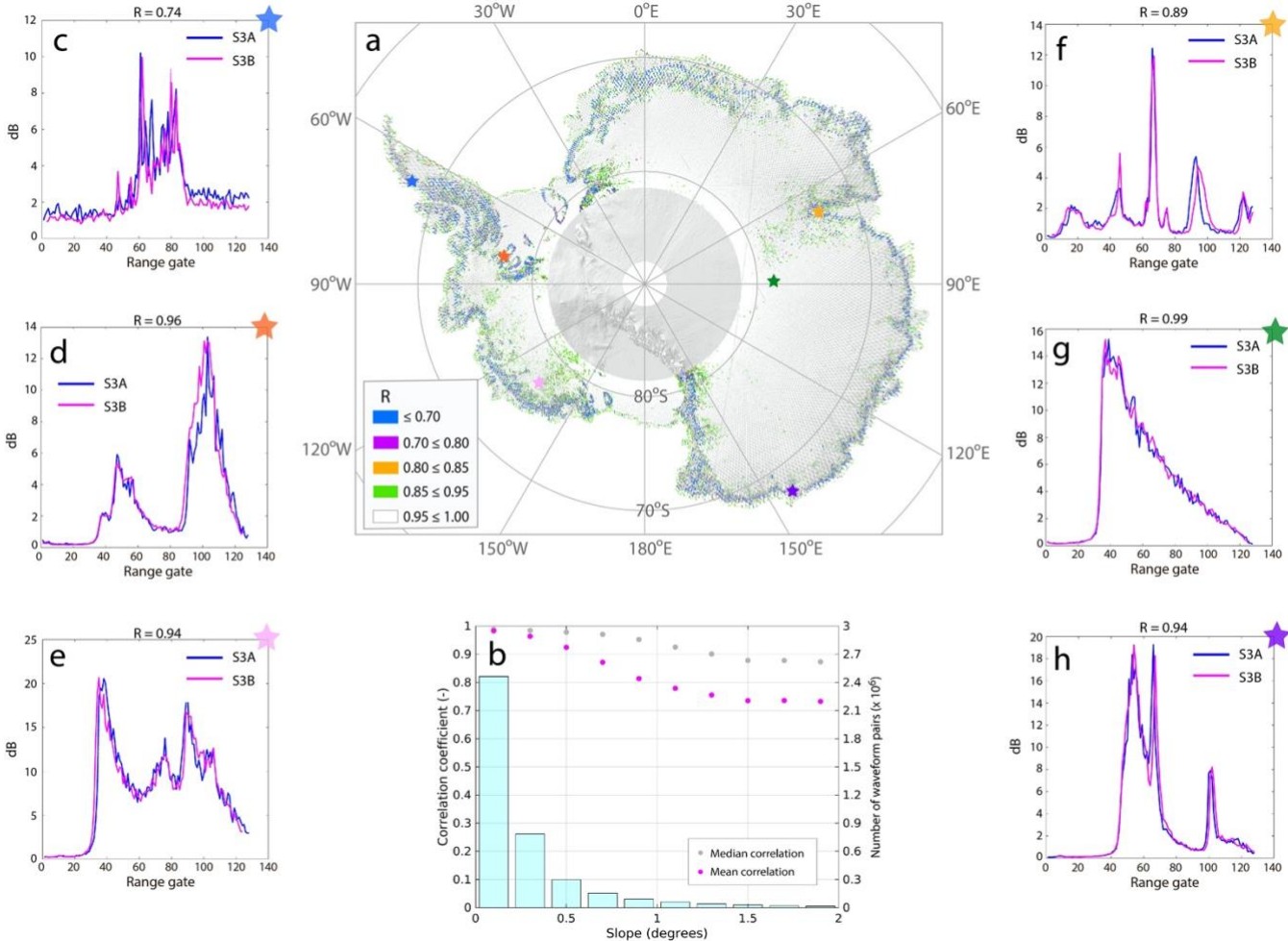

**Figure 1: The repeatability of Sentinel-3A (S3A) and Sentinel-3B (S3B) tandem acquisitions over the Antarctica Ice Sheet. Panel a. The Pearson correlation coefficient, R, for ~ 4 million S3A-S3B tandem waveform pairs; averaged on a 5 x 5 km grid. The coloured stars indicate the locations of the waveform pairs shown in panels c-h. Panel b. The mean (magenta) and median (grey) correlation coefficient of waveform pairs as a function of the gradient of surface slope. The turquoise bars show the number of waveform pairs averaged for each 0.2° slope interval. Panels c-h. Examples of S3A-S3B waveform pairs with varying levels of complexity. In panel a, the background image is a shaded relief of Antarctica (Slater et al., 2018), and the 5 x 5 km grid utilises a WGS84 Antarctic Polar Stereographic projection with 0°E central meridian and a 71°S Latitude of true origin.**

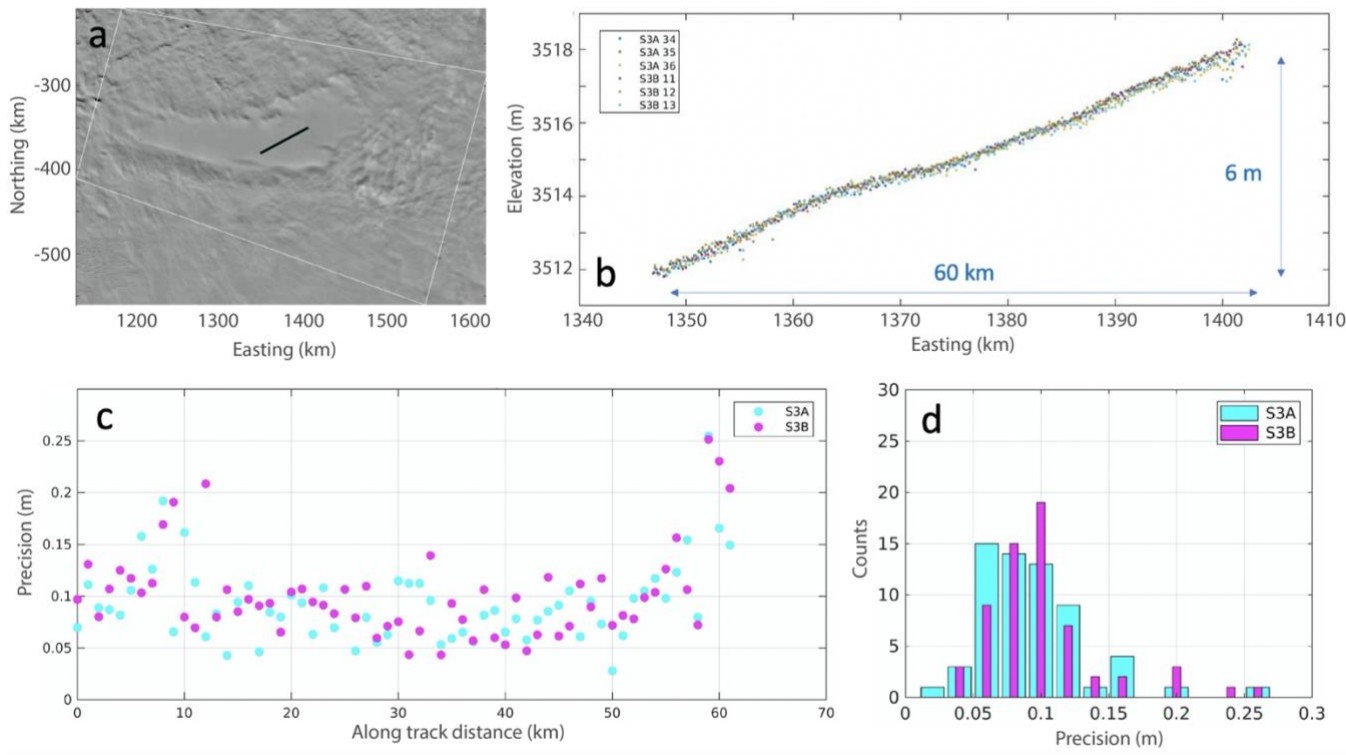

**Figure 2: Assessment of Sentinel-3A and Sentinel-3B instrument precision at Lake Vostok site. a. Location of the ground track crossing the central part of the lake (shown in black), which was used to assess precision; the background image is taken from the MODIS Mosaic of Antarctica (Haran et al., 2006). b. Repeated elevation profiles acquired during Sentinel-3A cycles 34-36 inclusive and Sentinel-3B cycles 11-13 inclusive. c. The standard deviation of Sentinel-3A and Sentinel-3B elevation measurements in 1 km intervals along the satellite track. d. The distribution of the 1 km-interval standard deviations for Sentienl-3A and Sentinel-3B; there is no significant difference in S3A and S3B precision at this site, at the 5% significance level. In panels a, b and c, the geographical coordinates refer to a WGS84 Antarctic Polar Stereographic projection with 0°E central meridian and a 71°S Latitude of true origin.**

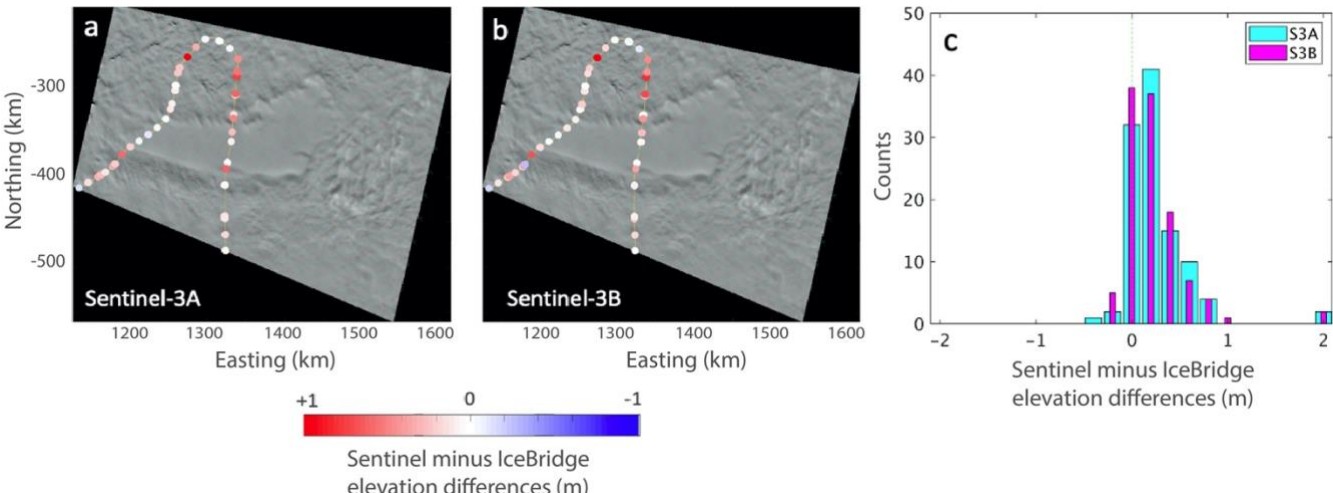

**Figure 3: Inter-comparison of Sentinel-3A and Sentinel-3B accuracy at the Lake Vostok Validation site. a. Sentinel-3A minus IceBridge elevation differences, b. Sentinel-3B minus IceBridge elevation differences, c. the distributions of Sentinel-3A and Sentinel-3B minus IceBridge elevation differences. Results shown are for the TCOG retracking solution provided within the ESA Level-2 product, and for acquisitions made during cycle 34 (S3A) and cycle 11 (S3B) within the tandem phase. The background image in panels a and b is taken from the MODIS Mosaic of Antarctica (Haran et al., 2006). In panels a and b, the geographical coordinates refer to a WGS84 Antarctic Polar Stereographic projection with 0°E central meridian and a 71°S Latitude of true origin.**