# Peer review of "Brief Communication: Ice Sheet Elevation Measurements from the Sentinel-3A / 3B Tandem Phase"

_The Cryosphere, 2020_

## Referee Comment (RC1) · Anonymous Referee #1 · 26 Sep 2020

This "Brief Communication" paper follows validations of the Sentinel-3 over Antarctic performed by McMillan et al. (2019). It uses observation data of the tandem phase of S3A and S3B to cross-check both missions. The study is well structured, very focused and nicely illustrated. It summarizes that there are no significant differences between S3A and S3B. In principle I agree with that statement, however, some of the analysis made do not totally convince me. In Sect. 3 I would have expected some more data being analyzed and in Sect. 5 I miss the numeric results that prove the statements made. Before publication, I suggest some modifications of the analyses as described below.

Specific comments:

l.38 If the across-track control range is ±1km, why are the shots co-located within ∼150m?

l.62 I think it is well known that surface topography is a major factor for conventional pulse-limited waveforms and the repeat measurements are supposed to match very well (especially in absence of other factors as temporal changes in surface properties or track orientation). Nevertheless, this comparison is very important to show how well the Doppler-stacking works in complex terrain. However, in my opinion, such an investigation should include a larger amount of data, not only 8 waveform examples of one track. As you have tandem data for more than one cycle, could you maybe find a quantity for "how well the waveforms of S3A and S3B match" (as maybe a correlation coefficient) and show this quantity on a larger scale (maybe even all over Antarctica)? This would prove that the stacking is reliable all over the ice sheet, not only for this specific location of this track. It would be also very interesting if there are places (e.g. in mountainous terrain) where the waveforms do not match.

Fig. 2a) I suggest using a color with more contrast for the track.

l.82 If I understand that right, this assessment combined all measurements from S3A and S3B (of the selected track) to calculate a mean elevation profile and then compared the measurements of each mission separately to that mean profile. This, however, would include a possible small offset between the missions as well and interpret it as random measurement noise. I suggest to do this assessment for the 3 tracks of each satellite separately. This allows to compare the precision of each mission and could also identify a possible offset in comparing the mean profiles.

l.93 Could you provide some numeric results for the non significant differences? This would be an important quantity for an estimation of how significant obtained results could be. Moreover, I understand that this "Brief Communication" is just an update to McMillan et al. 2019 and that the investigation should, therefore, be similar to

this paper. However, the only accuracy result shown here is a single OIB-profile from 2013 as a reference. In McMillan et al. 2019 this profile (I guess it is the same at Dome-C) is the only reference which is studied in the interior (only) and provides a standard deviation below several meters. How well is this profile itself quality checked? For Lake Vostok, there exists a wealth of independent in situ GNSS profiles (https://tc.copernicus.org/articles/11/1111/2017/), covering several years and cross-controlled at many crossover locations. I suggest considering these profiles for such an accuracy assessment.

---

## Referee Comment (RC2) · Anonymous Referee #2 · 24 Nov 2020

This brief communication (BC) provides an overview of what's gained by incorporating a tandem phase to the commissioning of new radar altimeters. As shown in the BC one could only have wished that this concept had been performed in the past during the commissioning of ERS-2, ENVISAT, and AltiKa. The BC limits the space available for more in-depth analysis of the data gathered in the tandem phase. However, after reading the manuscript I'm left with a feeling of wanting more. Here, a more classical cross-over analysis would have been obvious and a cross-over analysis of the two instruments during the tandem-phase might have given insights to the instrument degradation of S3A. However, I fully acknowledge that only the "added information" of the unique tandem phase is the scope of this BC. With this in mind, I'm only left with

minor comments to the BC.

Minor comments:

l11: Add Copernicus in front of Sentinel

l12: remove "each of"

l15: "co-located measurements to be acquired" to "co-located measurements of surface elevation, to be acquired"

l19: Shepherd 2019 should be 2020.

L24: add a reference to a technical paper about the program/Sentinel-3

L37: how many cycles?

L47: Suggest removing the section and incorporate the information in the intro.

L49: Baseline 2.27? Please elaborate on this baseline numbering, how does this compare to Baseline 004, which is available at the sci-hub?

L51: Spirit, please add the geographical information

L53-55: Please use the right references for these two data set as posted on NSIDC. I know the references are limited, but the big effort of collecting airborne data should be acknowledged.

L69: "complex topo..." should do. What is non-linear?

L71-72: coherent waveforms: Is this analyzed by eye or do you have a measure?

L73: "future possibilities": I can see how this is done for CS2 with the phase information, but how would you go about this here. Could you give some in-sights?

L79: "central part of..." Could you be more specific? I see the figure has coordinates, but what is the projection? The same is the case for Figures 1 and 3.

L85: the number of significant digits should be the same

L94: insignificant, I guess you used the same tests as above?

L99: "recent" the first

L104: What about the satellite degradation? S3A has now been in orbit 2 years more than S3B, is this what is seen in Figure 1 with a possibly noisier S3A waveform?

L105: "indicates" - This should be stated clearly that we would need similar observations to be made when S3C and S3D are entering service.

All figures are missing information about the projection and geographical coordinates.

Figure 1: For parts of the waveforms S3A looks noisier, is there a way to judge if this is the case? Add more information to section 3 about the inter-comparison of the waveforms.

Figure 3: suggests replacing a and b with an along-flight-trace profile.

---

## Author Comment (AC1) · 26 Feb 2021

We thank the reviewer for the time they have dedicated to reviewing our manuscript, and are grateful for their comments, which we believe have substantially improved the manuscript. We address each of the reviewer's comments in turn; reviewer's comments are prefixed by an asterisk.

Reviewer 1 Comments

\* This "Brief Communication" paper follows validations of the Sentinel-3 over Antarctic performed by McMillan et al. (2019). It uses observation data of the tandem phase of

[Figure]

S3A and S3B to cross-check both missions. The study is well structured, very focused and nicely illustrated. It summarizes that there are no significant differences between S3A and S3B. In principle I agree with that statement, however, some of the analysis made do not totally convince me. In Sect. 3 I would have expected some more data being analyzed and in Sect. 5 I miss the numeric results that prove the statements made. Before publication, I suggest some modifications of the analyses as described below.

As requested, we have now added significant additional content to both Sections 3 & 5. For Section 3, we have extended the analysis to a continent-wide evaluation, generating statistics for $\sim$ 4 million waveform pairs. We have also included a quantitative assessment of the relationship between waveform correlation and ice sheet surface slope. In addition to updates to the text, Figure 1 has been completely redrawn to reflect these new analyses. For Section 5, we have added more quantitative results. Further details are provided in response to the specific comments below.

Specific comments:

* l.38 If the across-track control range is $\pm$1km, why are the shots co-located within $\sim$150m?

We agree that our wording here was confusing. The control range is the *maximum* across track tolerance allowed. Usually, however, the separation is much less, and 150 m was the typical separation distance we found on the track analysed in our original Figure 1. We acknowledge that this was not at all clearly written and – as this text relates to the overall operation of the tandem phase, we have removed the reference to '150 m'. This is also more applicable now that we have expanded the analysis associated with Figure 1.

* l.62 I think it is well known that surface topography is a major factor for conventional pulse-limited waveforms and the repeat measurements are supposed to match very well (especially in absence of other factors as temporal changes in surface properties

or track orientation). Nevertheless, this comparison is very important to show how well the Doppler-stacking works in complex terrain. However, in my opinion, such an investigation should include a larger amount of data, not only 8 waveform examples of one track. As you have tandem data for more than one cycle, could you maybe find a quantity for "how well the waveforms of S3A and S3B match" (as maybe a correlation coefficient) and show this quantity on a larger scale (maybe even all over Antarctica)? This would prove that the stacking is reliable all over the ice sheet, not only for this specific location of this track. It would be also very interesting if there are places (e.g. in mountainous terrain) where the waveforms do not match.

We agree that this is an interesting and worthwhile analysis, and adds significantly to the results presented in the manuscript, so thank you for this suggestion. We have now undertaken a continental-scale analysis of $\sim$ 4 million S3A-S3B waveform pairs. We have computed the correlation between each pair, discussed the results within the text, and plotted the spatial distribution so the reader can see the performance in different sectors of the ice sheet. We have also undertaken a quantitative assessment of the relationship between the correlation coefficient and the surface slope of the ice sheet. The new results are presented in this section and also in a redrawn Figure 1.

* Fig. 2a) I suggest using a color with more contrast for the track.

As suggested, we have changed the colour of the track to one with greater contrast.

* l.82 If I understand that right, this assessment combined all measurements from S3A and S3B (of the selected track) to calculate a mean elevation profile and then compared the measurements of each mission separately to that mean profile. This, however, would include a possible small offset between the missions as well and interpret it as random measurement noise. I suggest to do this assessment for the 3 tracks of each satellite separately. This allows to compare the precision of each mission and could also identify a possible offset in comparing the mean profiles.

We agree with the reviewer, and we confirm that this is already the approach we have

taken, as described in the text: "We selected a track that crossed above the central part of the subglacial lake and, \*for each satellite\*, we accumulated consecutive cycles... we computed the standard deviation of all measurements \*made by each satellite\* within 1 km intervals along the satellite track".

\* l.93 Could you provide some numeric results for the non significant differences? This would be an important quantity for an estimation of how significant obtained results could be.

As requested, we have added such statistics to give the reader information relating to the magnitude of these differences.

\* Moreover, I understand that this "Brief Communication" is just an update to McMillan et al. 2019 and that the investigation should, therefore, be similar to this paper. However, the only accuracy result shown here is a single OIB-profile from 2013 as a reference. In McMillan et al. 2019 this profile (I guess it is the same at Dome-C) is the only reference which is studied in the interior (only) and provides a standard deviation below several meters. How well is this profile itself quality checked? For Lake Vostok, there exists a wealth of independent in situ GNSS profiles (https://tc.copernicus.org/articles/11/1111/2017/), covering several years and cross-controlled at many crossover locations. I suggest considering these profiles for such an accuracy assessment.

We agree that the GNSS profiles are, in principal, a very valuable dataset. However, for the purposes of this specific study, we would prefer to keep with the OIB comparison for the following reasons, (1) this is a Brief Communication (as the reviewer notes) and is therefore specifically designed to follow on from the analyses presented in McMillan et al., 2019 (and therefore for consistency we would prefer to keep the same validation datasets where possible), (2) here we are looking at \*relative\* differences between S3A and S3B, and as such the \*absolute\* accuracy of the validation dataset it not critical (as long as it is kept the same in both comparisons), and (3) the ground footprint of the

OIB ATM is closer to the resolution of the SAR footprint than a GNSS measurement, and therefore provides a closer like-for-like comparison (i.e. less sensitivity to small scale undulations in the surface topography than a point measurement).

---

## Author Comment (AC2) · 26 Feb 2021

We thank the reviewer for the time they have dedicated to reviewing our manuscript, and are grateful for their comments, which we believe have substantially improved the manuscript. We address each of the reviewer's comments in turn; reviewer's comments are prefixed by an asterisk.

* This brief communication (BC) provides an overview of what's gained by incorporating a tandem phase to the commissioning of new radar altimeters. As shown in the BC one could only have wished that this concept had been performed in the past during the commissioning of ERS-2, ENVISAT, and AltiKa. The BC limits the space available

for more in-depth analysis of the data gathered in the tandem phase. However, after reading the manuscript I'm left with a feeling of wanting more. Here, a more classical cross-over analysis would have been obvious and a cross-over analysis of the two instruments during the tandem-phase might have given insights to the instrument degradation of S3A. However, I fully acknowledge that only the "added information" of the unique tandem phase is the scope of this BC. With this in mind, I'm only left with minor comments to the BC.

As noted by the reviewer, we have chosen not to include a classical cross-over analysis here, because we wanted to focus on the specific novel aspects that could only be achieved thanks to the tandem phase. However, in our revised manuscript we have added considerable extra analysis and novel results that we hope will be of interest to Reviewer 2, and will satisfy their thirst to know more, within the scope of a Brief Communication.

Minor comments:

* l11: Add Copernicus in front of Sentinel

'Copernicus' added as requested.

* l12: remove "each of"

'each of' removed as requested.

* l15: "co-located measurements to be acquired" to "co-located measurements of surface elevation, to be acquired"

Text modified as requested.

* l19: Shepherd 2019 should be 2020.

This reference is, to our knowledge, correct as the paper was published on 16 May 2019: https://agupubs.onlinelibrary.wiley.com/doi/full/10.1029/2019GL082182.

* L24: add a reference to a technical paper about the program/Sentinel-3

Reference added as requested.

* L37: how many cycles?

Text added as requested: "three full cycles of tandem delay-Doppler measurements".

* L47: Suggest removing the section and incorporate the information in the intro.

We would prefer to keep this as a distinct section as we believe it is clearer for the reader.

* L49: Baseline 2.27? Please elaborate on this baseline numbering, how does this compare to Baseline 004, which is available at the sci-hub?

Baseline 2.27 relates the specific product baseline, which is part of a Baseline Collection 00X, e.g. '004' on sci-hub. In this case we used the Product Baseline 2.27, which is part of Baseline Collection 003, and we have clarified this point in the text.

* L51: Spirit, please add the geographical information.

Geographical coordinates added as requested.

* L53-55: Please use the right references for these two data set as posted on NSIDC. I know the references are limited, but the big effort of collecting airborne data should be acknowledged.

Apologies for this oversight – we have now added the correct references.

* L69: "complex topo..." should do. What is non-linear?

Text modified as requested. We had meant that elevation did not vary linearly with horizontal distance, but we agree that this wording was not clear and therefore unnecessary.

* L71-72: coherent waveforms: Is this analyzed by eye or do you have a measure?

It was the former in the original manuscript. However, please note that we have now undertaken a quantitative assessment, and also expanded it to the ice sheet scale – see our response to Reviewer 1's comments.

* L73: "future possibilities": I can see how this is done for CS2 with the phase information, but how would you go about this here. Could you give some in-sights?

As requested, we have elaborated on this point.

* L79: "central part of. . ." Could you be more specific? I see the figure has coordinates, but what is the projection? The same is the case for Figures 1 and 3.

As requested, we have added projection information to the captions of all figures. The projection used is the WGS 84 Antarctic Polar Stereographic projection with 0E central meridian and a 71S Latitude of true origin.

* L85: the number of significant digits should be the same

According to our understanding of significant figures, then the numbers 0.094 and 0.10 do have the same number of significant figures; i.e. 2. However, on reflection we think it would be better to include the same number of decimal places, and so we have updated accordingly.

* L94: insignificant, I guess you used the same tests as above?

Yes – text added to clarify this point.

* L99: "recent" the first

Text modified as suggested.

* L104: What about the satellite degradation? S3A has now been in orbit 2 years more than S3B, is this what is seen in Figure 1 with a possibly noisier S3A waveform?

For measurements of elevation, and elevation change, we believe that our analysis shows that satellite degradation over 2-years has had no significant impact upon the

elevation measurements themselves. In a further more technical study, it would be possible to look at whether there is any change in the noise statistics of the waveforms, both the multi-looked 20 Hz waveforms and also the lower level echoes. This would, however, require a far more detailed assessment of the lower level data than the 20 Hz comparison presented here. As such, we believe that this would be a distinct study, separate from this short communication.

* L105: "indicates" - This should be stated clearly that we would need similar observations to be made when S3C and S3D are entering service.

As suggested, we have modified the text so that this conclusion is more clearly stated.

* All figures are missing information about the projection and geographical coordinates.

As requested, we have added projection information to the captions of all figures. The projection used is the WGS 84 Antarctic Polar Stereographic projection with 0E central meridian and a 71S Latitude of true origin.

* Figure 1: For parts of the waveforms S3A looks noisier, is there a way to judge if this is the case? Add more information to section 3 about the inter-comparison of the waveforms.

As requested, we have now added considerably more detail relating to the inter-comparison of waveforms – please see our responses to Reviewer 1 and also the above comment relating to Line 104. In this study, we have chosen to focus on the overall correlation between the waveforms, because to perform an analysis of the noise content specifically would be a substantial technical piece of work, which is beyond the scope of this brief communication. For example, it would entail derivation, testing and analysis of methods that were able to separate instrument noise from topographic artefacts (i.e. real geophysical signals) within the waveform. We believe that this is well beyond the scope of this brief communication, and is a topic that should be considered as part of a future study.

\* Figure 3: suggests replacing a and b with an along-flight-trace profile.

We would prefer to keep the figure as is, because (1) it maintains consistency with McMillan et al., 2019, and (2) it more clearly shows the topographic configuration of the surface, via the MOA greyscale background image.

---

## Author Response (AR2)

Comments to the Author:
Dear authors,

Your paper has now been assessed again by one of the original referees, and I am happy to inform you that your manuscript is acceptable for publication once you have satisfactorily implemented the last few edits suggested by the referee (see below). These final edits will be assessed by me, before final publication.

Best,
Louise
* * *
referee comments:

I am pleased to see the much more convincing results in Sect.3 and acknowledge the additional work the authors have put into this analysis and several other parts of the manuscript. I have only a few remaining minor comments on the revised version.

One main point is that there are several places where the reader is referred to McMillan et al., 2019. This is okay when referring to details about the approach (as e.g. in l.111). However, Sect.5 is impossible to interpret at all without switching between those two papers several times. I think, also a "Brief Communication" should be understandable on its own (this refers mainly to my comments on l.104 and l.117).

Please see our individual responses to the comments below.

After the following minor comments are addressed, I suggest the manuscript for publication:

l.53 Why is this region called "Spirit site"? Isn't it Adélie Land / George V Land? How are OIB flights distributed there? You say, you want to make the validation comparable to McMillan et al., 2019, but why don't you use the regions "Dronning Maud Land" and "Wilkes Land" then?

Sorry, this was a poor choice of name – as highlighted by the reviewer – and we should have referred to it as 'Wilkes Land', as it is exactly the same area used in McMillan et al., 2019. The use of 'Spirit' stemmed from an informal name used in a previous project (*Sentinel-3 Performance Improvement for Ice Sheets*), but it is clearly more appropriate to use a more established Antarctica naming convention. We have therefore updated the text to address this oversight. Further details relating to the site can be found in McMillan et al., 2019.

Fig. 1a) I suggest to use a more gradual color scale like "Viridis" or "Plasma". Furthermore, it is difficult to decide if black color is the shade of the relief or means $R<=0.7$ (both would be expected in rugged terrain).

As suggested, we have adapted the colour scale to remove black and therefore avoid any confusion with the underlying greyscale shaded relief. Regarding the initial point, we have spent a long time trying to optimise the colour scale for clarity, including testing multiple variants of more gradual colour scales. However, we found that these were more difficult to

interpret in terms of being able to match a specific colour on the plot to a particular R value. Therefore, we would strongly prefer to keep a discrete colour scale, as we believe it is clearer and more easily interpretable for the reader.

l.104 I guess that "dispersion" means MAD. Please define (also for Sect. 5).

Here we are referring to the standard deviation, as already described in the text:

*'… we computed the standard deviation of all measurements made by each satellite within 1 km intervals along the satellite track. This yielded an estimate of the dispersion…'*

However, as this was not clear to the reviewer, we have reworded to try and add further clarity:

*'… we computed the standard deviation of all measurements made by each satellite within 1 km intervals along the satellite track. We used the estimated standard deviation as a measure of the precision…'*

As requested, we have also added further clarification relating to this point in Section 5.

l.111 Briefly mention what "reference datasets" are (OIB/airborne profiles, refer to Sect.2).

We have added text as requested.

l.117 In order to give the reader the ability to interpret the significance by her/his own, please also mention the median bias and the MAD between S3 and the reference dataset (or at least the order of magnitude). I know that this is given in Tab.3 of McMillan et al., 2019 but mentioning at least the key results (or order of magnitude) here would make this section understandable on its own.

As requested, we have added text here to address this point.